# 'Be on the TEAM' Study (Teenagers Against Meningitis): protocol for a controlled clinical trial evaluating the impact of 4CMenB or MenB-fHbp vaccination on the pharyngeal carriage of meningococci in adolescents

Jeremy Carr [1], Emma Plested,[1,2] Parvinder Aley,[1,2] Susana Camara,[1,2] Kimberly Davis,[1] Jenny M MacLennan,[3] Steve Gray,[4] Saul N Faust [5,6] Ray Borrow,[4] Hannah Christensen,[7] Caroline Trotter [8] Martin C J Maiden [3], Adam Finn [7] Matthew D Snape,[1,2] 'Be on the TEAM' investigators

For numbered affiliations see end of article.

**Correspondence to**
Dr Jeremy Carr;
jeremy.carr@paediatrics.ox.ac.uk

## ABSTRACT

**Introduction** Capsular group B *Neisseria meningitidis* (MenB) is the most common cause of invasive meningococcal disease (IMD) in many parts of the world. A MenB vaccine directed against the polysaccharide capsule remains elusive due to poor immunogenicity and safety concerns. The vaccines licensed for the prevention of MenB disease, 4CMenB (Bexsero) and MenB-fHbp (Trumenba), are serogroup B 'substitute' vaccines, comprised of subcapsular proteins and are designed to provide protection against most MenB disease-causing strains. In many high-income countries, such as the UK, adolescents are at increased risk of IMD and have the highest rates of meningococcal carriage. Beginning in the late 1990s, immunisation of this age group with the meningococcal group C conjugate vaccine reduced asymptomatic carriage and disrupted transmission of this organism, resulting in lower group C IMD incidence across all age groups. Whether vaccinating teenagers with the novel 'MenB' protein-based vaccines will prevent acquisition or reduce duration of carriage and generate herd protection was unknown at the time of vaccine introduction and could not be inferred from the effects of the conjugate vaccines. 4CMenB and MenB-fHbp may also impact on non-MenB disease-causing capsular groups as well as commensal *Neisseria* spp. This study will evaluate the impact of vaccination with 4CMenB or MenB-fHbp on oropharyngeal carriage of pathogenic meningococci in teenagers, and consequently the potential for these vaccines to provide broad community protection against MenB disease.

**Methods and analysis** The 'Be on the TEAM' (Teenagers Against Meningitis) Study is a pragmatic, partially randomised controlled trial of 24 000 students aged 16–19 years in their penultimate year of secondary school across the UK with regional allocation to a 0+6 month schedule of 4CMenB or MenB-fHbp or to a control group. Culture-confirmed oropharyngeal carriage will be assessed at baseline and at 12 months, following which

## Strengths and limitations of this study

► The scale of this carriage study will compensate for the known variation in carriage rates that can occur even when consistent methodology is used across multiple sites. The wide geographical distribution of study sites increases applicability to other settings.

► Regional allocation of schools to the same study group may increase the power to detect population effects in discrete social networks.

► The duration of post-immunisation follow-up (6 months) and a single post-immunisation sampling point limits the potential to identify a longer-term or more short-lived reduction in carriage, however, the mobility of school-leavers renders longer-term follow-up during the highest period of acquisition infeasible in this study.

► This study will inform modelling estimates of potential community-wide impact of adolescent immunisation with either 4CMenB or MenB-fHbp, which are dependent on whether these vaccines induce herd protection.

the control group will be eligible for 4CMenB vaccination. The primary outcome is the carriage prevalence of potentially pathogenic meningococci (defined as those with genogroups B, C, W, Y or X), in each vaccine group compared separately to the control group at 12 months post-enrolment, that is, 12 months after the first vaccine dose and 6 months after the second vaccine dose. Secondary outcomes include impact on carriage of: genogroup B meningococci; hyperinvasive meningococci; all meningococci; those meningococci expressing vaccine antigens and; other *Neisseria* spp. A sample size of 8000 in each arm will provide 80% power to detect a 30% reduction in meningococcal carriage, assuming genogroup B, C, W, Y or X meningococci carriage of 3.43%,

a design effect of 1.5, a retention rate of 80% and a significance level of 0.05. Study results will be available in 2021 and will inform the UK and international immunisation policy and future vaccine development.

**Ethics and dissemination** This study is approved by the National Health Service South Central Research Ethics Committee (18/SC/0055); the UK Health Research Authority (IRAS ID 239091) and the UK Medicines and Healthcare products Regulatory Agency. Publications arising from this study will be submitted to peer-reviewed journals. Study results will be disseminated in public forums, online, presented at local and international conferences and made available to the participating schools.

**Trial registration numbers** ISRCTN75858406; Pre-results, EudraCT 2017-004609-42.

## INTRODUCTION
### Background and rationale

Capsular group B *Neisseria meningitidis* (MenB) accounts for most cases of invasive meningococcal disease (IMD) in the UK, Europe, North America, Australia, New Zealand and many countries in Latin America.[1–3] Most IMD cases occur in infants and young children, with a smaller peak in incidence in late adolescence. Despite advances in recognition and management, the global case fatality rate remains 9%, with substantial sequelae in many survivors.[1] While conjugate vaccines that target the polysaccharide capsule of groups A, C, W and Y meningococci are highly effective, a capsule-based vaccine for MenB remains elusive due to poor immunogenicity and potential autoimmunity resulting from homology with surface glycoproteins on human embryonic neural cells.[4] Consequently the vaccines licensed for the prevention of IMD due to MenB, 4CMenB (Bexsero, GSK) and MenB-fHbp (Trumenba, Pfizer), target subcapsular proteins. These proteins are highly diverse and have variable density of surface expression. Thus, adequate breadth of coverage relies on a multi-component strategy and potential to generate cross-protective vaccine-induced antibodies. The 4CMenB vaccine contains factor H-binding protein (fHbp) from subfamily B/variant 1 fused with the accessory protein GNA2091; Neisserial heparin-binding antigen, Neisserial adhesin A fused with the accessory protein GNA1030 and an outer-membrane vesicle NZ98/254 (PorA 1.4). MenB-fHbp contains two self-lipidated alleles of fHbp, one each from subfamily A/variant 3 and subfamily B/variant 1. 4CMenB and MenB-fHbp were licensed based on safety and immunogenicity data determined by human serum bactericidal antibody activity as an established correlate of protection.[5 6]

In September 2015, the UK introduced a routine 4CMenB infant immunisation programme, in which all infants born after the 1st of May 2015 are eligible to receive a 2+1 schedule of 4CMenB at 2, 4 and 12 months of age. Ten months after introduction, the effectiveness of the 2-month and 4-month doses against all MenB IMD was estimated at 83% with very broad uncertainty (95% CI 24.1–95.2), a result consistent with the estimated breadth of coverage from immunogenicity studies.[3] At 3 years, expected cases of MenB IMD in vaccine-eligible children reduced by 75%.[7] Vaccine effectiveness of the 2+1 schedule was 59.1% (95% CI 31.1–97.2) against all MenB IMD and 71.2% against vaccine preventable MenB strains. There are no other vaccine effectiveness studies for either 4CMenB or MenB-fHbp.

Herd protection was not observed or expected from the UK MenB infant programme because carriage is very rare in infants as compared with the peak carriage rates that occur in late adolescence and early adulthood.[8] The potential for indirect protection to enhance vaccine impact and cost-effectiveness is exemplified by the meningococcal group C conjugate (MCC) vaccine campaign that included children, adolescents and young adults. A reduction in carriage in this targeted population dramatically reduced IMD caused by serogroup C meningococcus (MenC) in all age groups in the UK.[9] Maintaining this reduction in carriage and therefore MenC IMD in the UK is the primary rationale for an adolescent booster dose of a MenC containing vaccine in early adolescence following infant priming with a single MCC dose given at 12 months of age.

Based on modelling estimates, an infant-only MenB immunisation programme may prevent 26.3% of all-age IMD in the first 5 years after introduction.[10] If MenB vaccines could reduce meningococcal carriage by 30% and therefore disrupt transmission, a combined infant and adolescent immunisation programme could reduce all-age annual cases by 48.8% at 10 years and 59.7% at 20 years after introduction, enhancing impact and cost-effectiveness.[10] The impact of 4CMenB immunisation on carriage has been studied in two randomised controlled trials (RCTs) to date. The 'B Part of It' cluster-RCT assessed the impact of immunisation with 4CMenB on carriage in 24 269 secondary school students in Australia.[11] There was no impact on carriage of pathogenic genogroups A, B, C, W, Y or X at 12 months following immunisation. In the UK, 2954 university students were randomised to receive 4CMenB, MenACWY-CRM, or to an active control.[12] This study showed no effect on the primary outcome of carriage prevalence 1 month after a two-dose schedule of 4CMenB. Using a composite end point of carriage at any timepoint 3–12 months following immunisation, administration of 4CMenB reduced carriage of any meningococcus by 18.2% (95% CI 3.4–30.8) and meningococcus groups B, C, W and Y combined by 26.6% (95% CI 10.5–39.9). This reduction was primarily due to a fall in carriage of meningococci of capsular groups C, W and Y, with a non-significant change in capsular group B meningococcal carriage. Only a minority of students at any institution received the vaccine, which reduced the potential impact of herd protection in the cohort. Studies assessing MenB-fHbp carriage impact are limited to observational reports of several small MenB outbreaks in US universities,[13] where a number of interventions including mass-immunisation campaigns halted new cases, however these studies were not specifically designed or powered to show an impact of MenB-fHbp vaccine on carriage.

4CMenB and MenB-fHbp may elicit immune responses against non-MenB meningococci and, for 4CMenB, *N.*

*gonorrhoeae.*[14 15] Vaccine antigens may also be present in commensal *Neisseria* spp. resident in the oropharynx.[15 16] While cross-protection against meningococci expressing other disease-associated capsular groups is desirable, a broader impact on colonising strains of *N. meningitidis* or potentially protective non-invasive organisms, such as *N. lactamica,* requires surveillance. Unencapsulated meningococci, which are generally non-pathogenic, may have greater surface exposure of vaccine antigens and therefore be more vulnerable to mucosal clearance than the potentially invasive capsule-expressing meningococci. Natural exposure to these bacteria throughout childhood contributes to the development of immunity against hyperinvasive strains.[17] It is not known whether vaccines that induce immunity to these *Neisseria* spp. may facilitate an unintended niche for pathogenic meningococci.

## METHODS
### Study design
The 'Be on the TEAM' (Teenagers Against Meningitis) Study is a pragmatic, partially randomised cluster controlled trial that will evaluate the carriage prevalence of pathogenic meningococci in adolescents vaccinated with either 4CMenB or MenB-fHbp, compared with a control group. Participants aged 16–19 years will be enrolled in 16 regions across the UK, with each region allocated to: (1) vaccination at 0 and 6 months with 4CMenB ('4CMenB Group'); (2) vaccination at 0 and 6 months with MenB-fHbp ('MenB-fHbp Group') or (3) an unvaccinated Control Group who will be eligible for deferred vaccination with 4CMenB. Oropharyngeal swab samples (OPSs) will be taken at baseline and at 12 months.

### Primary Objective
▶ To determine the carriage prevalence of culture-confirmed *N. meningitidis* genogroup B, C, W, Y, X (ie, those meningococci possessing genes encoding the capsular polysaccharides corresponding to the respective serogroups) at 12 months after baseline vaccination (6 months following the second vaccine dose) in the 4CMenB and MenB-fHbp Groups compared with the unvaccinated Control Group. The study will not directly compare the 4CMenB Group with the MenB-fHbp Group.

### Secondary Objectives
To determine whether vaccination with either 4CMenB or MenB-fHbp affects the carriage prevalence of:
▶ Genogroup B meningococci.
▶ Meningococci of other genogroups and capsule null meningococci.
▶ Strains belonging to hyperinvasive meningococcal lineages.
▶ All meningococcal strains.
▶ Meningococci expressing antigens contained in MenB-fHbp or 4CMenB.
▶ Other *Neisseria* spp.

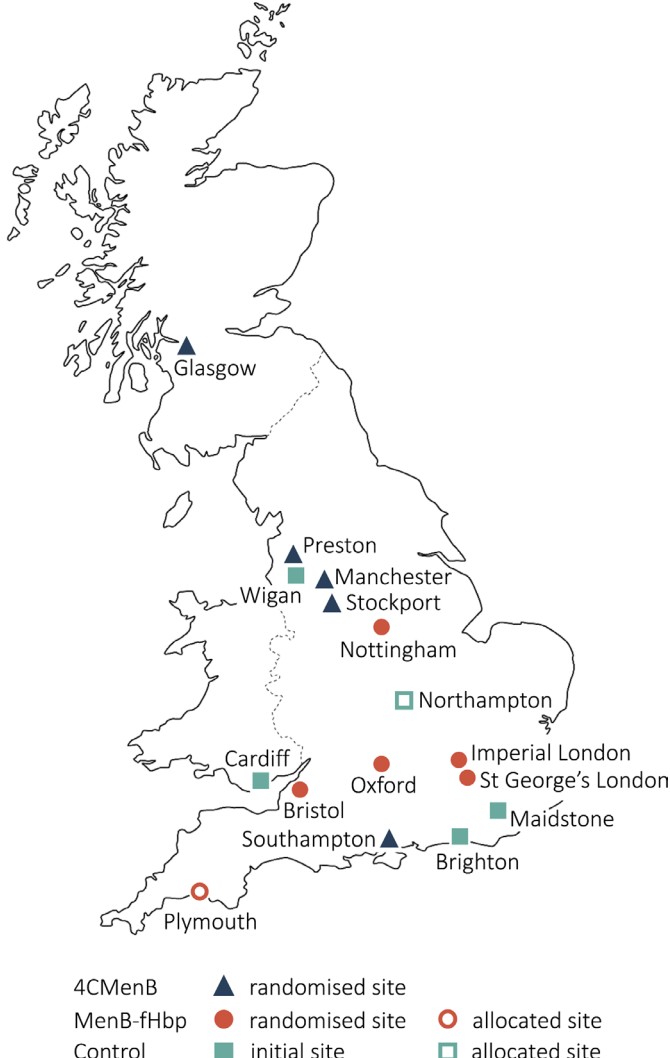

**Figure 1** Study sites and vaccine group allocation

A further secondary objective will be to determine any difference in rates of new acquisition of carriage between groups as defined in the primary and secondary objectives above.

As a stand-alone exploratory objective, this study will also permit an assessment of the impact of the UK adolescent quadrivalent MenACWY vaccination programme on carriage.[18] This campaign was introduced in 2015 due to a rise in capsular group W meningococcal disease and has high coverage in the school population of prospective participants in this study. 'UKMenCar4', an adolescent carriage study across the same study network in 2014–2015 will act as the pre-intervention study.[19 20] The post-intervention carriage prevalence will be determined by the baseline carriage samples from the Be on the TEAM Study.

### Study setting
This is a multi-centre, school-based study in 16 regions across the UK (figure 1). Students will be aged 16–19 years at time of enrolment, attending school or college year 12 (in England and Wales) or S5 (in Scotland), or

their equivalent. This will include both government (local authority) and independent schools, sixth form colleges, single sex or coeducation schools, boarding or day schools. Study visits will be conducted in schools, although follow-up visits may occur in a community setting if participants have left school.

## Sample size

The sample size of 24 000 participants (8000 in each arm) is based on an estimated carriage rate 3.43% for genogroup B, C, W, Y or X in the Control Group at the 12-month swab. When assuming a retention rate of 80% and a design effect of 1.5, this sample size will provide 80% power to detect a 30% reduction of these genogroups combined, at a significance level of 0.05.

This anticipated carriage rate is based on results from the UKMenCar4 Study, which determined a carriage prevalence of 1.65% for genogroup B meningococcus, 2.3% for hyperinvasive meningococci and a combined prevalence of 4.2% for genogroup B, C, W, Y and X meningococci.[19] The figure of 3.43% accounts for a potential 50% reduction in carriage of capsular groups W and Y due to the UK quadrivalent MenACWY vaccination programme; this figure is necessarily an estimate as there are limited data on the impact of the universal adolescent quadrivalent ACWY vaccine programme on carriage. A previous study in UK university students showed that immunisation with MenACWY-CRM reduced carriage of genogroup C, W or Y meningococci by 27.1% (95% CIs 6.9–42.9) and serogroup C, W or Y by 36.2% (95% CIs 15.6–51.7).[12] A further UK university carriage cohort study suggested that the quadrivalent MenACWY vaccine exerts differential effects on capsular groups and did not

prevent expansion of meningococci belonging to the hyperinvasive genogroup W clonal complex 11, although this was based on cross-sectional surveys rather than a longitudinal design.[21]

## Study timeline

In order to meet recruitment targets and maximise the interval between immunisation and the final OPS within the structure of the school exam year, the study will recruit across 4 'waves' over 20 months: wave 1 March–May 2018; wave 2 September–November 2018; wave 3 March–May 2019; wave 4 September–November 2019 (figure 2). The academic teaching year runs from September until April or May, depending on local jurisdictions. Recruitment will be balanced between study groups for each of the 'Spring' (March–May) and 'Autumn' (September–November) recruiting periods in view of expected seasonal variation in carriage. Completion of the final post-vaccination pharyngeal swab collection is expected in early 2021.

## Eligibility criteria

Participants will be healthy adolescents aged 16–19 years old at time of enrolment, in their penultimate year of participating secondary school or sixth form colleges, who intend to return for a further year of school. Students will be eligible for enrolment if they are willing and able to provide informed consent and meet the additional inclusion criteria:
► Willing to comply with all study requirements.
► Consent for storage of oropharyngeal swabs and any bacterial isolates for future research.

**Figure 2** Study timeline by recruitment wave and year 12/S5 cohort (penultimate year of secondary school) at the time of enrolment. Vaccine groups receive either MenB-fHbp or 4CMenB at 0+6 months. See figure 3 for visit window ranges.

► Willing to allow their general practitioner to be contacted to confirm vaccination status or medical history.

Participants will be ineligible to participate if they have received a course of either 4CMenB or MenB-fHbp in the past (by documentation or self-report). Furthermore, participants in the 4CMenB Group or MenB-fHbp Group will be ineligible if they meet any of the following criteria:

1. History of anaphylaxis to any component of 4CMenB or MenB-fHbp (For 4CMenB only, this includes allergy to latex).
2. Any other significant disease or disorder, which in the opinion of the investigator may: put the participants at risk because of participation in the trial, influence the result of the trial or affect the participant's ability to participate. Specific examples include haemophilia or medically diagnosed bleeding disorder, or receipt of anticoagulant medication, that prohibits the use of intramuscular injections.
3. Known or suspected pregnancy.

Participants in the Control Group that fail any of the specific exclusion criteria 1, 2 or 3 will be eligible to participate in the assessment of carriage, but will not be able to receive the 4CMenB vaccine after the final OPS. Ineligibility arising during the trial due to criteria 1, 2 or 3 in the 4CMenB Group or MenB-fHbp Group will allow inclusion for assessment of carriage at the study endpoint but will exclude a participant from vaccination.

## STUDY PROCEDURES
### Recruitment
Recruitment will occur in participating schools; individual participants will not be directly recruited by the study teams outside of the school setting. Social media and the study website (www.beontheteam.uk) will be used for study information and awareness, but these platforms will not be used for recruitment. Participants will receive information through a school assembly about meningococcal disease and the study rationale and design. Subsequently, they will receive a written participant information sheet.

### Informed consent
All participants will be aged 16 years or over, and are therefore self-consenting as per the National Institute of Health Research guidelines.[22] However, parents/guardians will be provided with information about the study via school communication platforms as well as directly by prospective participants. Consent will be taken by clinical staff (registered doctor or nurse) or non-clinical staff who have had appropriate experience and training. Where interested participants have vulnerabilities that may impair their capacity to provide informed consent, additional input or support from the individual's parents/guardians will be sought. If there is ongoing doubt about an individual's ability to provide informed consent, then they will not be enrolled in the study.

### Randomisation, allocation and blinding
This is an open-label partially randomised study, with group allocation by study site. All schools within the geographical coverage area of each study site will be allocated to the same study arm, both for logistical reasons and because geographical allocation may increase the potential to detect any impact on herd protection. As a post-licensure study, blinded administration and/or placebo will not impact on the outcome of carriage prevalence and would introduce unnecessary cost and complexity to a trial of this scale. The allocation of initial sites was determined prior to the start of the study (figure 1). A capacity assessment stratified sites to either the control arm or immunisation sites depending on existing site expertise and the requirements to maintain equal numbers in each study arm. The immunisation sites were randomised to either the MenB-fHbp Group or the 4CMenB Group. Additional sites may be added during the study and will be pragmatically allocated to a study group to ensure balanced recruitment.

### Interventions
All participants will be required to attend three study visits at their school (figure 3). Following informed consent, a baseline OPS will be taken by study staff using a standardised method of collection. A flocked swab will be used to sweep from one tonsil across the posterior pharynx to the contralateral tonsil. Participants in all groups will complete a brief questionnaire collecting demographic data and risk factors for meningococcal carriage such as smoking and antimicrobial use (see online supplemental file).[23] The questionnaire is compatible with previous large-scale UK meningococcal carriage studies conducted in 1999, 2000, 2001 and 2014–2015, and the B Part of It Australian Study, which will allow an assessment of risk factors over time.[11 19 24 25] The 4CMenB and MenB-fHbp groups will both receive a two-dose vaccine schedule at 0 and 6 months and all groups will have a second OPS taken at 12 months following enrolment. The Control Group will be offered a two-dose schedule of 4CMenB vaccination after their second OPS. Individual swab results will not be given to participants, in accordance with previous meningococcal carriage studies, however participants will be contacted confidentially if *N. gonorrhoeae* is detected.

### Patient and public involvement
This study will provide a unique opportunity to engage adolescents on an unprecedented scale to: (1) increase awareness of meningococcal disease; (2) highlight the importance of immunisation and (3) stimulate an interest in involvement in impact-driven research that potentially benefits the broader community. All year 12 (or equivalent) students at participating schools will attend assemblies, often involving meningitis survivors and disease-prevention advocates. These 'meningitis ambassadors' will discuss their personal experience of meningitis and provide education about signs and symptoms of meningitis. An ethically approved presentation

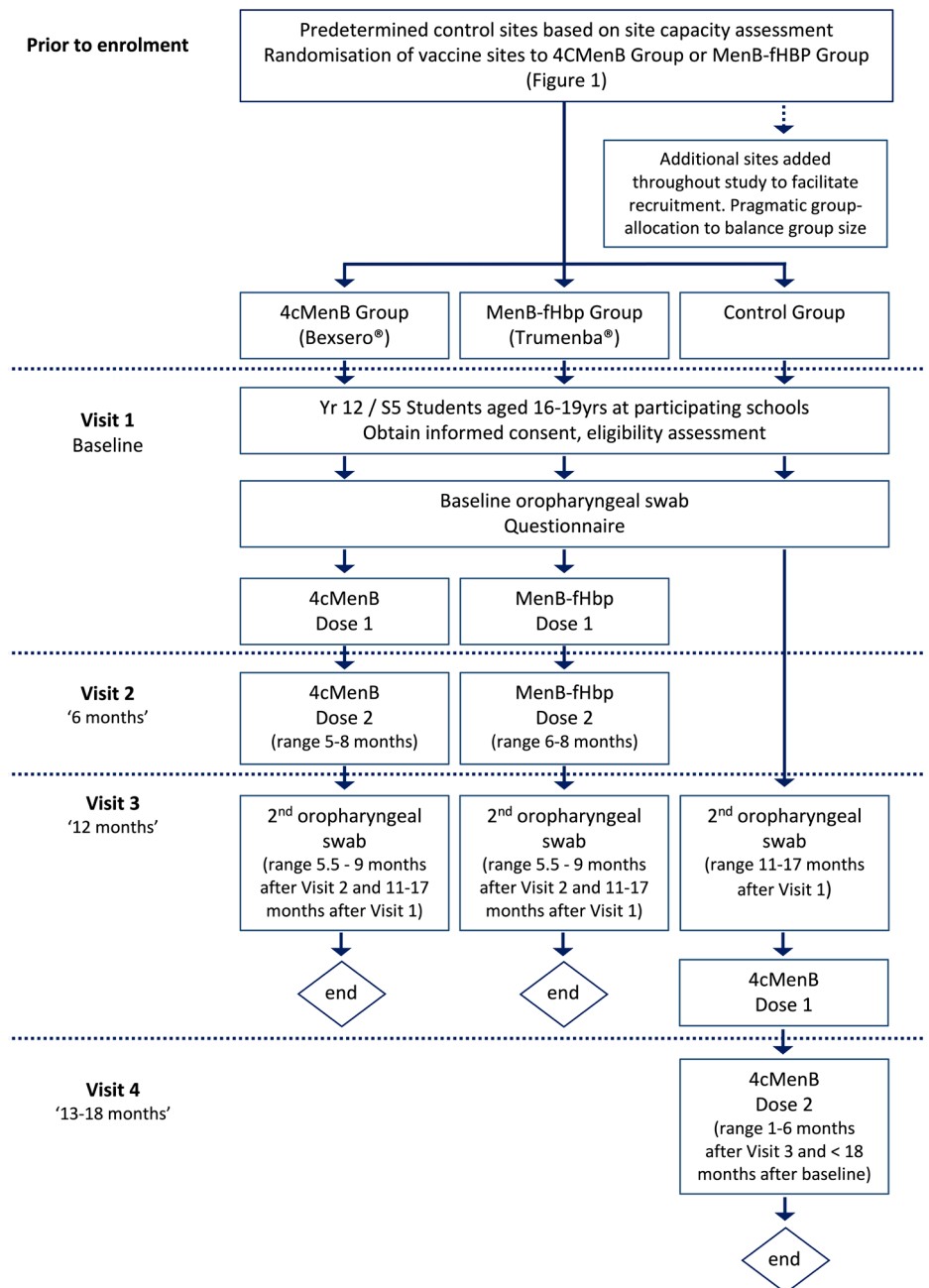

**Figure 3** Study design

will describe the science behind the study, including information about IMD and the link between vaccination, carriage and herd protection. Communication material used in this study will be developed with input from focus groups run by two meningitis charities, Meningitis Now and the Meningitis Research Foundation, who will also coordinate the provision of 'meningitis ambassadors'. Other initiatives include a social media policy engaging students, schools and research teams with curated social media posts to complement public engagement aims. Combining these strategies will ensure that participants make a fully informed decision to participate in the study. Furthermore, by involving entire school year cohorts

(regardless of enrolment into the study), an estimated 48 000 students across 200 schools will be engaged in a critical conversation about meningitis, research and the benefits of immunisation across the broader community. Study results will be submitted for publication in peer-reviewed journals, and then disseminated directly to the schools involved, as well as on the study social media platforms, and the Meningitis Now and Meningitis Research Foundation websites.

### Equality and diversity
The study will be done across 16 regions throughout the UK and will include schools across the range of

 Carr J, *et al. BMJ Open* 2020;**10**:e037358. doi:10.1136/bmjopen-2020-037358

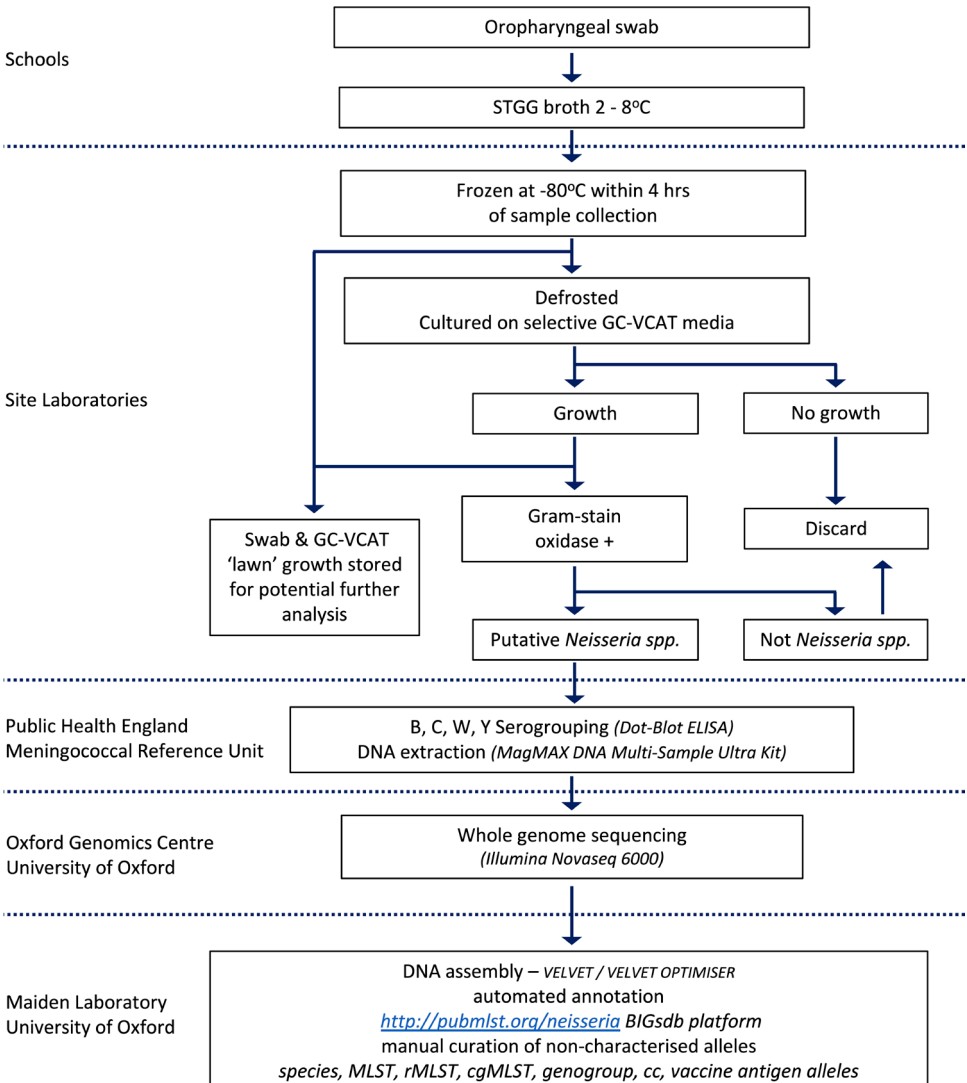

**Figure 4** Laboratory analysis plan. CC clonal complex; cgMLST, core genome MLST; GC-VCAT, vancomycin, colistin, amphotericin B, trimethoprim; MLST, multilocus sequence typing; rMLST, ribosomal MLST; STGG; skim milk, tryptone, glucose, glycerine.

socioeconomic strata. School types reflect the variety of educational institutions across the UK as detailed in the study setting. Previous diversity information collected as part of UKMenCar4 across the same study network demonstrated that participants were reflective of their communities according to national census data.[20] Inclusion of whole schools, rather than targeting individuals, will encourage equal participation of diverse populations.

### Sample handling and laboratory processes

Collected OPS will be placed immediately into STGG broth (skim milk, tryptone, glucose, glycerine; EO Labs, Bonnybridge, UK), maintained at 2°C–8°C until frozen at −80°C, within 4 hours of collection (figure 4). Site laboratories will thaw and plate the OPS sample on selective GC-VCAT Agar (vancomycin, colistin amphotericin B, trimethoprim; Thermo Fisher Scientific, Basingstoke, UK) at 37°C in 5% $CO_2$ for up to 48 hours. Putative *Neisseria* spp. (oxidase positive, Gram-negative diplococci) will be subcultured on to Columbia Horse Blood Agar,

then stored in a bead-based cryovial (Technical Service Consultants, Lancashire, UK) at −80°C.

The Public Health England Meningococcal Reference Unit (Manchester) will serogroup putative *Neisseria* spp. isolates using an in-house dot-blot ELISA using National Institute for Biological Standards and Control monoclonal antibodies against capsular polysaccharide serogroups B, C, W and Y.[26] DNA will be extracted using a bead-based MagMAX DNA Multi-Sample Ultra Kit (Thermo Fisher Scientific, Massachusetts, USA). Total DNA extracts will be normalised prior to whole genome sequencing on the Illumina NovaSeq6000 platform (Oxford Genomic Centre, Wellcome Trust Centre for Human Genetics, University of Oxford).

Genomes will be assembled using an in-house assembly pipeline (Department of Zoology, University of Oxford) which incorporates Velvet and VelvetOptimiser.[27 28] Assembled draft genome sequences will then be submitted to the open-access online database http://pubMLST.org/

neisseria, with isolate providence data and FASTQ files uploaded to the European Nucleotide Archive and the Sequence Read Archive Databases. Automated locus annotation using the BIGSdb software platform will be used concurrently with manual curation for classification of isolates including species, clonal complex, lineage, genogroup and meningococcal typing schemes for multilocus sequence typing (MLST), ribosomal MLST, core genome MLST and vaccine antigen variants (eg, Bexsero Antigen Sequence Type).[29 30] After study completion, genomes will be published open access on the pubMLST.org/neisseria database and OPSs, *Neisseria* spp. isolates and any products of culture on GC-VCAT media will be stored in an approved facility for further research.

### Potential benefits and harms

Participants will benefit from immunisation with a licensed 'MenB' vaccine which they would not otherwise receive in the UK immunisation schedule. Prior to enrolment, participants will receive information about 4CMenB and MenB-fHbp vaccination, according to the licensed product information. As a participant engagement exercise, all participants completing the final swabbing visit will be entered into a prize-draw for headphones. Participants will not receive payments.

### Safety reporting and study oversight

This study involves products used according to their licences. Safety monitoring will focus on detecting any serious adverse events (SAEs) and suspected unexpected serious adverse reactions (SUSARs). Specific SAE exemptions include traumatic injuries, hospital admissions for intoxication, deliberate self-harm or elective management of pre-existing conditions unless in response to a worsening of that condition since study enrolment. All SAEs will be reported to the chief investigator and to the sponsor, the University of Oxford Clinical Trials and Research Group. The Oxford University Hospitals National Health Service Foundation Trust/University of Oxford Trials Safety Group will review all new SAEs on a quarterly basis. SUSARs will be reported by the study chief investigator to the relevant competent authority and to the Research Ethics Committee. The lead site is the Oxford Vaccine Group, Department of Paediatrics, University of Oxford. The Scientific Advisory Board will provide input into study design, statistical analysis and will oversee safety reporting and study conduct in accordance with Good Clinical Practice. The Oxford Vaccine Group on behalf of the sponsor will act as the study monitor.

### Statistical methods

A baseline analysis will occur after wave 2 when the study reaches approximately 50% enrolment. This descriptive analysis of baseline carriage rates will assess the validity of the prospective sample size calculation. There will be no interim analysis of vaccine impact. The principal analysis will be conducted using a modified intention to treat approach, including all participants who have had two throat swabs. Participants will be analysed according to allocated group, regardless of vaccine receipt. Based on previous UK school-based meningococcal carriage studies, the design effect was assumed to be 1.5 in the sample size calculations but will be calculated on the actual data collected.[24 25]

For the primary objective and each of the secondary objectives, carriage rates will be expressed as a mean with two-sided 95% Clopper-Pearson CIs for each group at baseline and at 12 months. The carriage rates in the MenB-fHbp and 4CMenB groups will be compared separately to the Control Group. The OR with two-sided 95% CIs will be calculated using logistic regression via a generalised estimating equations model using an exchangeable working correlation structure. Model covariates will include age, gender, season of recruitment and baseline carriage status a priori. Additional risk factors found to affect carriage at baseline will also be included. Gender and smoking (smoking, home smoking and vaping) are known to affect both vaccine-induced immune responses and meningococcal carriage, and these interactions will be reported regardless of significance. The level of statistical significance will be 0.05. There will be no imputation of missing data for the principal analysis.

Sensitivity analyses will be done for the primary and each of the secondary objectives and will include: analyses without adjustment or inclusion of covariates for completeness and to confirm the results from the primary analyses; a per-protocol analysis of participants who received both vaccine doses in the 4CMenB and MenB-fHbp groups; adjustment for the month of the outcome measure rather than season of recruitment to further investigate the potential effect of seasonality and multivariable imputation for missing data. All variables that will be included in the analysis model will be included in the imputation model and will assume data are missing at random.

### Data collection and management

Informed consent forms will be completed on paper. Source data entry will be on a standard case report form, either paper based, or direct electronic entry using REDCap, a browser-based electronic data capture record, hosted by the University of Oxford.[31] Participant data will be handled in accordance with Good Clinical Practice and the General Data Protection Act. Participants will consent to the storage of de-identified OPSs, isolates and questionnaire data. All identifiable data will be destroyed after the youngest participant in the trial turns 21 years.

### DISCUSSION

This is the first large-scale adolescent meningococcal carriage study to assess the impact of both the 4CMenB and MenB-fHbp vaccines on carriage of pathogenic meningococci as well as commensal *Neisseria* spp. Building on an established meningococcal carriage and vaccine clinical trial network, the backward compatibility of this

study will allow comparisons with four previous carriage studies in 1999, 2000, 2001 and 2014–2015. This study will also facilitate an evaluation of the UK adolescent quadrivalent MenACWY vaccination programme on carriage since its introduction in 2015. The 16 sites across the UK represent a geographically and demographically diverse population, increasing applicability to broad settings.

The 'B Part of It' trial showed that there is no significant difference in carriage of genogroup A, B, C, W, Y and X meningococci between adolescents who received 4CMenB compared with controls.[11] Overall, the carriage of genogroup B meningococcus was low. The 'B Part of It' Study has a similar design as a school-based study of adolescents aged 16–18 years; however, there are some notable difference between the design of the 'B Part of It' Study and the present study, specifically, the 'Be on the TEAM' Study:

1. Includes both MenB-fHbp and 4CMenB.
2. Allocates vaccine arms on a regional basis rather than cluster randomisation at the level of schools, which may increase the potential to observe herd protection.
3. Uses a 0-month and 6-month vaccine schedule, longer than the 1–2 month interval used in the 'B Part of It' Study.
4. Has a culture-defined endpoint of carriage, compared with direct PCR amplification targeting PorA in the 'B Part of It' Study.

Despite the difference in study designs, cross-study collaborations will enrich understanding of the potential vaccine impact on specific *N. meningitidis* strains and commensal *Neisseria* spp. by genome level interrogation.

The short follow-up period of between 6 and 9 months following the second vaccination is a limitation of this study, reducing the ability to observe any potential impact during later periods of high-acquisition risk, for example, when commencing tertiary education.[32] The large sample size required and mobility of school-leavers limit the feasibility of ongoing carriage surveillance. The secondary endpoint assessing the impact on other *Neisseria* spp. is based on those present in the oropharynx and culturable on selective media, however all swabs and products of culture will be stored for potential further analysis.

Due for completion in 2021, The 'Be on the TEAM' Study will directly inform international MenB immunisation policy. It will contribute to an open-access repository of *Neisseria* spp. genomes and provide a biobank of samples to assist in identifying strategies for future MenB vaccine development.

## Current study progress

At the time of writing 24 048 participants have been enrolled from over 120 schools across the UK. Assemblies raising awareness about meningitis, immunisation and research have reached over 40 000 students. In March 2020, study visits were temporarily suspended due to the SARS-CoV-2 pandemic and at this time, 11 170 participants had successfully completed the study. The protocol may require modifications to fulfil the study objectives.

## Protocol version

Current Protocol Version 5.0 dated 24-Jan-2020. All protocol modifications will be reported promptly to the NIHR, Department of Health and Human Services, the Scientific Advisory Board and the Principal Investigators. This report was written using the SPIRIT checklist.[33] The study protocol, patient information sheets and an example consent form are publicly available on the study website: www.beontheteam.uk.

**Author affiliations**
[1]Oxford Vaccine Group, Department of Paediatrics, University of Oxford, Oxford, UK
[2]National Institute for Health Research Oxford Biomedical Research Centre, Oxford, UK
[3]Department of Zoology, University of Oxford, Oxford, UK
[4]Meningococcal Reference Unit, Public Health England, Manchester Royal Infirmary, Manchester, UK
[5]NIHR Southampton Clinical Research Facility and NIHR Southampton Biomedical Research Centre, University of Southampton and University Hospital Southampton NHS Foundation Trust, Southampton, UK
[6]Faculty of Medicine and Institute of Life Sciences, University of Southampton, Southampton, UK
[7]School of Population Health Sciences, Bristol Medical School, University of Bristol, Bristol, UK
[8]Department of Veterinary Medicine, University of Cambridge, Cambridge, UK

**Acknowledgements** The Scientific Advisory Board: Chair—Sir Brian Greenwood; Deputy Chair—James Stuart; Simon Nadel; Helen Marshall; Helen Bedford; Nick Andrews. Holly Bratcher and Odile Harrison, Department of Zoology, University of Oxford; Melanie Carr, Karen Ford, Hannah Roberts, Yama Farooq, Simon Kerridge and Annabel Giddings, Department of Paediatrics, University of Oxford; Shari Sapuan, St George's University NHS Foundation Trust; Rebecca Ramsay Brighton and Sussex University Hospitals NHS Foundation Trust; Dr Katrina Cathie, NIHR Southampton Clinical Research Facility and Department of Paediatrics, University Hospital Southampton NHS Foundation Trust. The authors acknowledge Meningitis Now and Meningitis Research Foundation for their input into study branding and design, educational material and organising meningitis ambassadors.

**Collaborators** 'Be on the TEAM' investigators: Andrew Smith, Christopher Williams, Christos Zipitis, Claire Cameron, David Baxter, David Orr, David Turner, Elizabeth Whittaker, Katy Fidler, Mala Raman, Paul Heath, Rohit Gowda, Stephen Hughes, Sujata Khajuria.

**Contributors** The study protocol was developed by MS, AF, MM, CT, HC, RB, SG, JM, PH, SF, KD, SC, PA, EP, JM, JC. CT and HC provided statistical expertise. The manuscript was drafted by JC and MS with input and editing from all authors and collaborators. All authors and collaborators reviewed the final manuscript and approved its contents.

**Funding** This study is independent research funded by the National Institute for Health Research (NIHR) Policy Research Programme ('Be on the TEAM', 'Evaluating the effect of 'MenB' vaccination on meningococcal carriage', PR-R18-0117-21001). Matthew Snape and the Oxford Vaccine Group, Department of Paediatrics, are supported by the NIHR Oxford Biomedical Research Centre. MenB-fHbp donated by Pfizer. Meningitis Now provided funding for a prize draw to facilitate participant retention.

**Disclaimer** The views expressed in this publication are those of the authors and not necessarily those of the NHS, the NIHR or the Department of Health and Social Care. The funders have no role in the study design, conduct, analysis or publications arising from this study.

**Map disclaimer** The depiction of boundaries on this map does not imply the expression of any opinion whatsoever on the part of BMJ (or any member of its group) concerning the legal status of any country, territory, jurisdiction or area or of its authorities. This map is provided without any warranty of any kind, either express or implied.

**Competing interests** AF reports grants from the Department of Health, during the conduct of the study; grants from Pfizer, grants from GSK, outside the submitted work. HC reports other from Sanofi Pasteur, other from IMS Health, grants from National Institute for Health Research, other from AstraZeneca, grants and other

from GSK, outside the submitted work. MS reports grants from National Institute for Health Research (NIHR), non-financial support from Pfizer, during the conduct of the study; grants from Pfizer, grants from MCM, grants from GlaxoSmithKline, grants from Novavax, grants from Medimmune, grants from Janssen, outside the submitted work. SC reports salary contribution from the study NIHR grant, during the conduct of the study. SF reports personal fees were paid to his institution (with no personal payment of any kind) from AstraZeneca/Medimmune, Sanofi, Pfizer, Seqirus, Sandoz, Merck, GSK, Alios, Johnson & Johnson and Merck, outside the submitted work. RB reports grants from University of Oxford via a NIHR grant, during the conduct of the study; contract research on behalf of Public Health England for GSK, Pfizer and Sanofi Pasteur, outside the submitted work.

**Patient and public involvement** Patients and/or the public were involved in the design, or conduct, or reporting, or dissemination plans of this research. Refer to the Methods section for further details.

**Patient consent for publication** Not required.

**Provenance and peer review** Not commissioned; externally peer reviewed.

**ORCID iDs**
Jeremy Carr http://orcid.org/0000-0002-2729-0920
Saul N Faust http://orcid.org/0000-0003-3410-7642
Caroline Trotter http://orcid.org/0000-0003-4000-2708
Martin C J Maiden http://orcid.org/0000-0001-6321-5138
Adam Finn http://orcid.org/0000-0003-1756-5668

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
