## [Reviewer comments · BMJ Open]

ARTICLE DETAILS

TITLE (PROVISIONAL)	The 'Be on the TEAM' Study (Teenagers Against Meningitis): Protocol for a controlled clinical trial evaluating the impact of 4CMenB or MenB-fHbp vaccination on the pharyngeal carriage of meningococci in adolescents.
AUTHORS	Carr, Jeremy; Plested, Emma; Aley, Parvinder; Camara, Susana; Davis, Kimberly; MacLennan, Jenny; Gray, Steve; Faust, Saul; Borrow, Ray; Christensen, Hannah; Trotter, Caroline; Maiden, Martin; Finn, Adam; Snape, Matthew

VERSION 1 – REVIEW

REVIEWER	Dr Sarah Blagden Public Health England, United Kingdom
REVIEW RETURNED	26-Feb-2020

GENERAL COMMENTS	This is a really interesting protocol and I look forward to reading the results of the study when complete. I have a couple of minor points: 1) The questionnaire asks participants if they have received MenACWY vaccination and the recruitment of participants through school assemblies and discussion in this format will most likely stimulate widespread discussion (and possibly concern) about meningococcal disease and meningitis. Where students did not receive MenACWY for whatever reason, was information provided to them that they were eligible for catch-up vaccination and how to receive this? 2) Participants are recruited from their penultimate year of secondary school or sixth form college. Therefore, although they were recruited from a range of geographical areas, the sample may be skewed towards higher socioeconomic groups with higher education status as those students that left school in year 11 to go on to apprenticeships, work-based training etc will be excluded. This should be mentioned as a limitation and possibly accounted for through the statistical methods, especially as meningococcal carriage is associated with deprivation due to factors such as smoking prevalence.
--

REVIEWER	Patrick Schnell The Ohio State University, United States
REVIEW RETURNED	07-Apr-2020

GENERAL COMMENTS	This is a large, well-designed study from a statistical standpoint with a potential to have high impact. It has apparently already completed its large-scale enrollment, which would have seemed
--

	like a potential hurdle at the outset. I suggest minor revisions for clarity.  1. It is not difficult to hypothesize about why the trial was open label vs blinded and placebo-controlled, but an explicit justification for the decision would be helpful. 2. Please briefly explain how the recruiting waves align with terms or significant landmark dates on the academic calendar, for readers not familiar with the local school system. 3. For the power calculation, what is the justification for assuming a design effect of 1.5? 4. In the analysis plan, please specify the planned working correlation structure for the GEE. Is it, e.g., exchangeable within regions, or hierarchical with schools within regions?
--	--

VERSION 1 – AUTHOR RESPONSE

Reviewer: 1

Reviewer Name: Dr Sarah Blagden

Institution and Country: Public Health England, United Kingdom

Please state any competing interests or state 'None declared': None declared

This is a really interesting protocol and I look forward to reading the results of the study when complete.

I have a couple of minor points:

1) The questionnaire asks participants if they have received MenACWY vaccination and the recruitment of participants through school assemblies and discussion in this format will most likely stimulate widespread discussion (and possibly concern) about meningococcal disease and meningitis. Where students did not receive MenACWY for whatever reason, was information provided to them that they were eligible for catch-up vaccination and how to receive this?

The participants in this study were eligible for school-based vaccinations approximately 1-3 years prior to recruitment in the study. This school-based programme has high levels of coverage. Participants were able to contact their local study team for information and questions about any aspect of the study. The study website also has a public facing FAQ section that discussed MenB and MenACWY immunisation. School nursing teams / pastoral care are involved in the study visits and for the opportunity to ask questions about school immunisations.

2) Participants are recruited from their penultimate year of secondary school or sixth form college. Therefore, although they were recruited from a range of geographical areas, the sample may be skewed towards higher socioeconomic groups with higher education status as those students that left school in year 11 to go on to apprenticeships, work-based training etc will be excluded. This should be mentioned as a limitation and possibly accounted for through the statistical methods, especially as meningococcal carriage is associated with deprivation due to factors such as smoking prevalence.

We agree that including only enrolled school or college students is a practical limitation of the study, in terms of the applicability of the specific meningococcal carriage prevalence to the general population. This is less of a concern as an outcome measure for vaccine impact on carriage because this study is a comparison of carriage rates between similar school/college-based cohort in the control and vaccine groups, rather than a carriage prevalence survey representative of the true population. Variables such as smoking (including household smoking) will be included in the adjusted odds ratios of carriage comparisons between the control and vaccine groups. Carriage may be higher in the non-school groups, with relevance to the sample size calculations, however the sample size calculations related to a similar school-based survey in 2015.

Reviewer: 2

Reviewer Name: Patrick Schnell

Institution and Country: The Ohio State University, United States

Please state any competing interests or state 'None declared': None declared

This is a large, well-designed study from a statistical standpoint with a potential to have high impact. It has apparently already completed its large-scale enrollment, which would have seemed like a potential hurdle at the outset. I suggest minor revisions for clarity.

1. It is not difficult to hypothesize about why the trial was open label vs blinded and placebo-controlled, but an explicit justification for the decision would be helpful.

Detail added to 'Randomisation, Allocation and Blinding' section. As a post-licensure, large scale study, blinded administration and/or a placebo is unlikely have an impact on the outcome of (blinded) carriage prevalence and would introduce significant cost and complexity to a trial of this scale. Group allocation to one of the three study arms by Site/region increased potential to detect indirect protection and this would not be possible with blinded or placebo-controlled study designs.

2. Please briefly explain how the recruiting waves align with terms or significant landmark dates on the academic calendar, for readers not familiar with the local school system.

Explanations for the start of the new school year and terms added.

3. For the power calculation, what is the justification for assuming a design effect of 1.5?

References added. This is based on the calculated design effect for previous UK school-based meningococcal carriage studies across a similar geographic region (data provided to lead statistician Caroline Trotter).

4. In the analysis plan, please specify the planned working correlation structure for the GEE. Is it, e.g., exchangeable within regions, or hierarchical with schools within regions?

We will use an exchangeable working correlation structure (added to manuscript). Further details will be included in the stand-alone statistical analysis plan which will be made publicly available alongside study results.

VERSION 2 – REVIEW

REVIEWER	Patrick Schnell The Ohio State University College of Public Health, United States
REVIEW RETURNED	22-Jul-2020
GENERAL COMMENTS	The authors have addressed all of my recommendations in the previous round.